# Lipid Emulsions Inhibit Labetalol-Induced Vasodilation in the Isolated Rat Aorta

**DOI:** 10.3390/ijms25137243

**Published:** 2024-06-30

**Authors:** Soohee Lee, Kyeong-Eon Park, Yeran Hwang, Sungil Bae, Seong-Ho Ok, Seung-Hyun Ahn, Gyujin Sim, Hyun-Jin Kim, Seunghyeon Park, Ju-Tae Sohn

**Affiliations:** 1Department of Anesthesiology and Pain Medicine, Gyeongsang National University Changwon Hospital, Changwon-si 51472, Gyeongsangnam-do, Republic of Koreamdoksh@naver.com (S.-H.O.); 2Department of Anesthesiology and Pain Medicine, Gyeongsang National University College of Medicine, Jinju-si 52727, Gyeongsangnam-do, Republic of Korea; 3Institute of Medical Science, Gyeongsang National University, Jinju-si 52727, Gyeongsangnam-do, Republic of Korea; 4Department of Anesthesiology and Pain Medicine, Gyeongsang National University College of Medicine, Gyeongsang National University Hospital, 15 Jinju-daero 816 Beon-gil, Jinju-si 52727, Gyeongsangnam-do, Republic of Koreababi9876@naver.com (Y.H.); 5Department of Anesthesiology and Pain Medicine, Gyeongsang National University Hospital, 15 Jinju-daero 816 Beon-gil, Jinju-si 52727, Gyeongsangnam-do, Republic of Korea; 6Division of Applied Life Science (BK21 Four), Gyeonsang National University, Jinju-si 52828, Gyeongsangnam-do, Republic of Korea; 7Department of Food Science and Technology, Institute of Agriculture and Life Science, Gyeongsang National University, Jinju-si 52828, Gyeongsangnam-do, Republic of Korea

**Keywords:** endothelial nitric oxide, labetalol, lipid emulsion, nitric oxide, vasodilation

## Abstract

Lipid emulsions are used as adjuvant drugs to alleviate intractable cardiovascular collapse induced by drug toxicity. We aimed to examine the effect of lipid emulsions on labetalol-induced vasodilation and the underlying mechanism in the isolated rat aorta. We studied the effects of endothelial denudation, N^W^-nitro-l-arginine methyl ester (l-NAME), calmidazolium, methylene blue, 1*H*-[1,2,4]oxadiazolo[4,3-a] quinoxalin-1-one (ODQ), and lipid emulsions on labetalol-induced vasodilation. We also evaluated the effects of lipid emulsions on cyclic guanosine monophosphate (cGMP) formation, endothelial nitric oxide synthase (eNOS) phosphorylation, and endothelial calcium levels induced by labetalol. Labetalol-induced vasodilation was higher in endothelium-intact aortas than that in endothelium-denuded aortas. l-NAME, calmidazolium, methylene blue, and ODQ inhibited labetalol-induced vasodilation in endothelium-intact aortas. Lipid emulsions inhibited labetalol-induced vasodilation in endothelium-intact and endothelium-denuded aortas. l-NAME, ODQ, and lipid emulsions inhibited labetalol-induced cGMP formation in endothelium-intact aortas. Lipid emulsions reversed the stimulatory and inhibitory eNOS (Ser1177 and Thr495) phosphorylation induced by labetalol in human umbilical vein endothelial cells and inhibited the labetalol-induced endothelial calcium increase. Moreover, it decreased labetalol concentration. These results suggest that lipid emulsions inhibit vasodilation induced by toxic doses of labetalol, which is mediated by the inhibition of endothelial nitric oxide release and reduction of labetalol concentration.

## 1. Introduction

Labetalol, a non-selective beta and selective alpha-1 adrenoceptor blocker, is used to treat hypertension [1]. It is also used to treat acute hypertension during pregnancy [2]. Pre-treatment with the nitric oxide synthase (NOS) inhibitor N^G^-nitro-l-arginine methyl ester does not significantly reduce labetalol-induced vasodilation in the human radial artery, suggesting that labetalol-induced vasodilation is independent of endothelial nitric oxide release [3]. However, the referenced study did not confirm the endothelial integrity of the radial artery obtained from patients undergoing coronary artery bypass grafting. Thus, it is difficult to determine whether the insignificant reduction in labetalol-induced vasodilation by N^G^-nitro-l-arginine methyl ester was due to endothelial nitric oxide-independent vasodilation or compromised endothelial dysfunction of the radial artery obtained from patients undergoing coronary artery bypass grafting. Additionally, labetalol elicits greater relaxation in the endothelium-intact aorta compared to the in endothelium-denuded aorta, suggesting that labetalol-induced vasodilation is endothelium-dependent [4]. Although endothelial nitric oxide is involved in the regulation of vascular tone and blood flow, its role in labetalol-induced vasodilation remains unknown. 

Specific pharmacological treatments for labetalol toxicity include calcium, glucagon, catecholamines, insulin-euglycemic therapy, and sodium bicarbonate [5]. In addition, lipid emulsions, originally developed for parenteral nutrition, were reported to alleviate intractable hemodynamic depression caused by toxic doses of labetalol and amlodipine, which are resistant to supportive treatment [6]. Recently, lipid emulsions, as adjuvant drugs, have been reported to treat cardiovascular collapse caused by toxic doses of non-local anesthetic drugs with high lipid solubility (log P = log [octanol/water] partition coefficient > 2) [7,8,9,10,11]. Lipid emulsion resuscitation involves both indirect and direct mechanisms [9]. The lipid shuttle mechanism is a widely accepted indirect mechanism [9]. The lipid shuttle mechanism states that lipid emulsions absorb lipid-soluble drugs (e.g., bupivacaine, log P: 3.41) from the heart and brain, and lipid emulsions with lipid-soluble drugs are transported to the liver, muscle, and adipose tissue for detoxification and storage [9]. In addition, lipid emulsions attenuate vasodilation induced by the calcium channel blocker amlodipine, adenosine triphosphate (ATP)-sensitive potassium channel agonist levcromakalim, and acetylcholine, which is mediated by the inhibition of nitric oxide release [12,13,14]. However, the effect of lipid emulsions on vasodilation induced by toxic doses of the highly lipid-soluble labetalol (log P = 2.7) remains unknown [15]. Therefore, based on previous reports [9,12,13,14], we tested the biological hypothesis that lipid emulsions reverse the vasodilation induced by toxic doses of labetalol. The primary goal of this study was to examine the role of the endothelium in labetalol-induced vasodilation in isolated rat aortas, with a particular focus on endothelial nitric oxide. Additionally, we aimed to investigate the effect of lipid emulsions (Intralipid) on vasodilation induced by toxic doses of labetalol in isolated rat aortas and elucidate the associated mechanism. 

## 2. Results

### 2.1. Effect of Endothelial Denudation, Various Inhibitors, Preconstrictors, and Lipid Emulsion on Labetalol-Induced Vasodilation in Isolated Rat Aortas 

Labetalol-induced vasodilation was significantly higher in the endothelium-intact aortas compared to that in the endothelium-denuded aortas (Figure 1A,B; *p* < 0.001 versus the endothelium-denuded aortas at 3 × 10^−7^ to 3 × 10^−5^ M). The log ED_50_ (logarithm of labetalol concentration needed to produce half of the labetalol-induced maximal relaxation) also differed significantly between the two groups (*p* < 0.001: −6.11 ± 0.11 for endothelium-intact aortas versus −5.20 ± 0.15 for endothelium-denuded aortas). Pre-treatment with the NOS inhibitor N^W^-nitro-l-arginine methyl ester (l-NAME, 10^−4^ M) significantly attenuated labetalol-induced vasodilation in endothelium-intact aortas (Figure 1C; *p* < 0.01, versus control at 3 × 10^−7^ to 3 × 10^−5^ M). The log ED_50_ values also were significantly different (*p* < 0.001: −6.01 ± 0.18 for the control group versus −5.39 ± 0.17 for the l-NAME group). The calmodulin-regulated enzyme inhibitor calmidazolium (3 × 10^−5^ M) inhibited labetalol-induced vasodilation in endothelium-intact aortas (Figure 2A; *p* < 0.05 versus control at 3 × 10^−7^ to 3 × 10^−4^ M). Additionally, the log ED_50_ values were significantly different (*p* < 0.001: −5.95 ± 0.23 for the control group versus −5.57 ± 0.12 for the calmidazolium-treated group). Pre-treatment with the non-specific guanylate cyclase (GC) inhibitor methylene blue (10^−6^ M) or the nitric oxide-sensitive GC inhibitor 1*H*-[1,2,4]oxadiazolo [4,3-a] quinoxalin-1-one (ODQ, 10^−5^ M) inhibited labetalol-induced vasodilation in endothelium-intact aortas (Figure 2B,C; *p* < 0.001 versus the control at 3 × 10^−7^ to 10^−5^ M). The log ED_50_ values were also significantly different (*p* < 0.001: −6.11 ± 0.15 for control versus −5.56 ± 0.22 for methylene blue and −5.92 ± 0.30 for control versus −5.34 ± 0.14 for ODQ). The ATP-sensitive potassium channel inhibitor glibenclamide (5 × 10^−6^ M) had no effect on labetalol-induced vasodilation in endothelium-intact aortas (Figure 3A). Precontraction with phenylephrine (10^−6^ M) produced greater labetalol-induced vasodilation than that with 60 mM KCl or 5-hydroxytryptamine (3 × 10^−6^ M) in endothelium-denuded aortas (*p* < 0.001 versus 5-hydroxytryptamine or KCl at 10^−6^ to 3 × 10^−4^ M labetalol; Figure 3B). Lipid emulsion (1%) inhibited labetalol-induced vasodilation in endothelium-intact aortas without or with l-NAME (10^−4^ M) (Figure 4A,B; *p* < 0.001 versus the control or l-NAME alone at 10^−6^ and 3 × 10^−6^ M). Additionally, lipid emulsions (1%) inhibited labetalol-induced vasodilation in the endothelium-denuded aortas (Figure 4C; *p* < 0.05 versus the control at 10^−5^ and 3 × 10^−5^ M). The difference in the area under the labetalol concentration–response curve between the control and 1% lipid emulsion groups was greater in the endothelium-intact aortas than in the endothelium-denuded aortas (Figure 5A; *p* < 0.01).

### 2.2. Effect of Lipid Emulsion on Labetalol Concentration in Distilled Water

The lipid emulsions (1%) reduced labetalol (10^−5^ M) concentration (Figure 5B; *p* < 0.001 versus labetalol alone).

### 2.3. Effect of Labetalol, Inhibitor, and Lipid Emulsions, Alone or Combined, on Cyclic Guanosine Monophosphate (cGMP) Formation 

Labetalol (3 × 10^−6^ M) increased cGMP formation in isolated endothelium-intact aortas (Figure 6A,B; *p* < 0.001 vs. the control). l-NAME (10^−4^ M) and ODQ (10^−5^ M) inhibited labetalol (3 × 10^−6^ M)-induced cGMP formation in isolated endothelium-intact aortas (Figure 6A; *p* < 0.001 vs. labetalol alone). Additionally, the lipid emulsions (1%) attenuated labetalol (3 × 10^−6^ M)-induced cGMP formation in isolated endothelium-intact aortas (Figure 6B; *p* < 0.001 vs. labetalol alone). 

### 2.4. Effect of Labetalol and Lipid Emulsion, Alone or Combined, on Endothelial NOS (eNOS) Phosphorylation in Human Umbilical Vein Endothelial Cells (HUVECs)

Labetalol (3 × 10^−6^ M) increased stimulatory eNOS (Ser1177) phosphorylation in HUVECs (Figure 7A; *p* < 0.001 vs. control), whereas labetalol (3 × 10^−6^ M) decreased inhibitory eNOS (Thr495) phosphorylation in HUVECs (Figure 7B; *p* < 0.001 vs. control). However, lipid emulsion (1%) reversed the stimulatory eNOS (Ser1177) and inhibitory eNOS (Thr495) phosphorylation induced by labetalol in HUVECs (Figure 7A; *p* < 0.001 versus labetalol alone at eNOS [Ser1177]; Figure 7B; *p* < 0.05 versus labetalol alone at eNOS [Thr495]).

### 2.5. Effect of Labetalol, Inhibitor, and Lipid Emulsions, Alone or Combined, on Calcium Level in HUVECs

Labetalol (3 × 10^−6^ M) increased the intracellular calcium levels in HUVECs (Figure 8). However, pre-treatment with l-NAME (10^−4^ M) or lipid emulsions (1%) attenuated the increase in the calcium levels induced by 3 × 10^−6^ M labetalol in HUVECs (Figure 8; *p* < 0.001 versus labetalol alone).

## 3. Discussion

This study suggests that lipid emulsion attenuates vasodilation induced by a toxic dose of labetalol in isolated rat aortas, mediated through the inhibition of nitric oxide release and reduction of labetalol concentration. The major findings of this in vitro study were as follows: (1) labetalol-induced vasodilation was greater in endothelium-intact aortas than that in endothelium-denuded aortas; (2) l-NAME, ODQ, and methylene blue inhibited labetalol-induced vasodilation and labetalol-induced cGMP formation in the endothelium-intact aorta; (3) lipid emulsion inhibited labetalol-induced vasodilation in endothelium-intact and endothelium-denuded aortas; (4) lipid emulsions reversed stimulatory and inhibitory eNOS phosphorylation induced by labetalol, and inhibited labetalol-induced endothelial calcium increase.

Endothelial nitric oxide is produced from l-arginine by eNOS, which binds to calmodulin in a calcium-dependent manner [16,17]. Then, nitric oxide diffuses from the endothelium into the vascular smooth muscle [17]. Nitric oxide activates GC, leading to the formation of cGMP [17]. Subsequently, cGMP activates cGMP-dependent protein kinases leading to vasodilation [17]. Consistent with a previous report, labetalol-induced vasodilation was higher in the endothelium-intact aortas than that in the endothelium-denuded aortas, suggesting that labetalol-induced vasodilation is endothelium-dependent [4]. Thus, we examined the effects of various inhibitors of pathways associated with endothelial nitric oxide-induced vasodilation on labetalol-induced vasodilation in endothelium-intact aortas. The NOS inhibitor l-NAME and calmodulin-regulated enzyme inhibitor calmidazolium attenuated labetalol-induced vasodilation in endothelium-intact aortas, suggesting that labetalol-induced vasodilation is partially mediated by endothelial nitric oxide. In addition, the non-specific GC inhibitor methylene blue and the nitric oxide-sensitive GC inhibitor ODQ attenuated labetalol-induced vasodilation. However, pre-treatment with the NOS inhibitor N^G^-nitro-l-arginine methyl ester does not significantly reduce labetalol-induced vasodilation in the human radial artery [3]. This discrepancy between our study and previous reports may be attributed to differences in the experimental methods (confirmation vs. non-confirmation of endothelial integrity), species, and vessel location. One reason for this difference may be the compromised endothelium of the radial artery obtained from patients undergoing coronary artery bypass graft [3]. In agreement with the results of the tension study, labetalol increased cGMP production in endothelium-intact aortas, whereas ODQ and l-NAME decreased labetalol-induced cGMP production (Figure 6A). To the best of our knowledge, this is the first study to demonstrate that labetalol-induced endothelium-dependent vasodilation is mediated by a pathway involving endothelial nitric oxide-GC-cGMP. Vasodilation induced by ATP-sensitive potassium channels is partially mediated by endothelial nitric oxide [18]. Thus, we examined the effect of the ATP-sensitive potassium channel inhibitor glibenclamide on labetalol-induced vasodilation in the endothelium-intact aortas. The pre-treatment with glibenclamide had no effect on labetalol-induced vasodilation in the endothelium-intact aorta (Figure 3A), suggesting that labetalol-induced nitric oxide-mediated vasodilation does not involve the activation of ATP-sensitive potassium channels. Labetalol inhibits the pressure response to phenylephrine and decreases mean blood pressure in conscious, spontaneously-hypertensive rats [19]. Consistent with the findings of previous in vivo and in vitro studies, the magnitude of labetalol-induced vasodilation in endothelium-denuded aortas was greater in the aorta pre-contracted with the alpha-1 adrenoceptor agonist phenylephrine than in those pre-contracted with the receptor-mediated contractile agonist 5-hydroxytryptamine or voltage-operated calcium channel agonist 60 mM KCl (Figure 3B) [4,19]. This result indicates that labetalol-induced vasodilation in endothelium-denuded aortas is mediated mainly by alpha-1 adrenoceptor blockade.

Lipid emulsions alone increase blood pressure (systolic and diastolic) and vascular resistance, and reduce flow-mediated vasodilation and arterial compliance [20,21]. Moreover, lipid emulsions inhibit acetylcholine-induced nitric oxide-mediated vasodilation and phosphorylation of eNOS (Ser1177) [12]. These previous reports suggested that increased blood pressure and vascular resistance, as well as reduced flow-mediated vasodilation, which are observed after lipid emulsion administration, could be attributed to lipid emulsion-induced inhibition of nitric oxide release. Similar to previous reports, lipid emulsions inhibited labetalol-induced vasodilation and cGMP production (Figure 6B) in endothelium-intact aortas, suggesting that lipid emulsions have an inhibitory effect on nitric oxide release, which contributes to endothelium-dependent vasodilation by labetalol [12,20,21]. In addition, the lipid emulsions inhibited labetalol-induced vasodilation in endothelium-denuded and endothelium-intact aortas pre-treated with the NOS inhibitor l-NAME. These results indicate that lipid emulsions inhibit endothelial nitric oxide-independent vasodilation induced by labetalol. Lipid emulsions inhibited labetalol-induced vasodilation at lower concentrations (3 × 10^−7^ to 3 × 10^−6^ M) of labetalol in the endothelium-intact aorta. This inhibition could be attributed to both lipid emulsion-induced inhibition of labetalol-induced nitric oxide release and lipid emulsion-mediated sequestration of labetalol, which seems to be effective at low concentrations of labetalol (Figure 4A). Moreover, lipid emulsions inhibited labetalol-induced vasodilation at relatively high concentrations of labetalol (10^−5^ and 3 × 10^−5^ M) in endothelium-denuded aortas (Figure 4C), which can be attributed to lipid emulsion-mediated sequestration of labetalol. As lipid emulsion inhibited labetalol-induced vasodilation in both endothelium-intact and endothelium-denuded rat aortas, we investigated the involvement of inhibited nitric oxide release in lipid emulsion-mediated inhibition of labetalol-induced vasodilation in endothelium-intact aortas by comparing the difference in area under the labetalol-induced concentration–response curve in both cases, in the presence or absence of lipid emulsions (1%). Taken together, the difference in the area under the labetalol-induced concentration–response curve with or without lipid emulsions (1%) was greater in the endothelium-intact aortas than in the endothelium-denuded aortas (Figure 5A), suggesting that the lipid emulsion-mediated inhibition of nitric oxide release partially contributes to the lipid emulsion-mediated inhibition of labetalol-induced vasodilation in endothelium-intact aortas. As lipid emulsion decreased labetalol concentration (10^−5^ M) (Figure 5B) and labetalol is highly lipid-soluble (log P of labetalol: 2.7), lipid emulsion-mediated inhibition of labetalol-induced vasodilation in endothelium-denuded and endothelium-intact aortas with l-NAME may be attributed to the absorption (scavenging) of lipid-soluble labetalol by the lipid emulsions [7,15]. The mechanism underlying lipid emulsion resuscitation as an adjuvant drug for intractable drug toxicity involves both direct and indirect effects [9]. The direct effect of lipid emulsion resuscitation involves a positive inotropic effect, fatty acid supply, attenuation of mitochondrial dysfunction, attenuation of nitric oxide release, and glycogen synthase kinase-3β phosphorylation [9]. However, as lipid emulsions reduced labetalol concentration in the current study, it is difficult to distinguish lipid emulsion-induced absorption of labetalol (indirect effect) from lipid emulsion-induced direct inhibition of labetalol-induced nitric oxide production (direct effect) in understanding which mechanism contributes to the lipid emulsion-mediated inhibition of labetalol-induced vasodilation. Therefore, further studies employing experimental methods to distinguish the direct and indirect effects of lipid emulsions on labetalol-induced vasodilation are required.

Ser1177 phosphorylation stimulates eNOS, leading to increased eNOS activity and nitric oxide production, whereas Thr495 phosphorylation inhibits eNOS, leading to decreased eNOS activity and nitric oxide production [22]. Consistent with the tension study, labetalol increased the phosphorylation of eNOS (Ser1177) and decreased that of inhibitory eNOS (Thr495) (Figure 7A,B), leading to increased nitric oxide production. However, lipid emulsions (1%) reversed the stimulatory eNOS (Ser1177) phosphorylation and inhibitory eNOS (Thr495) dephosphorylation induced by labetalol, which led to a decrease in nitric oxide production. Additional research is needed to investigate the upstream cellular signaling pathways that induce stimulatory eNOS (Ser1177) phosphorylation and inhibitory eNOS (Thr495) dephosphorylation induced by labetalol. Calcium increase in endothelial cells is essential for activating calcium-dependent eNOS by calmodulin and subsequently producing nitric oxide [22]. Consistent with eNOS activation, nitric oxide-dependent vasodilation, and increased cGMP production, which were induced by labetalol in the current study, labetalol increased intracellular calcium levels in HUVECs. However, pre-treatment with l-NAME and lipid emulsions reduced the labetalol-induced calcium increase in HUVECs, likely leading to decreased nitric oxide production, as confirmed by the tension study and western blot analysis results. Acetylcholine produces intracellular calcium oscillations associated with nitric oxide release in endothelial cells, which is induced by endogenous calcium release from the endoplasmic reticulum and calcium entry via store-operated calcium channels [23]. Nitric oxide inhibits endothelial calcium levels via a cGMP-dependent mechanism, but also increases endothelial calcium through a cGMP-independent mechanism [24]. Moreover, nitric oxide decreases endothelial calcium levels by attenuating capacitative calcium entry and enhancing calcium uptake into the endoplasmic reticulum via a cGMP-dependent mechanism [25]. This contributes to the autoregulation of nitric oxide production [25]. Further studies are needed to investigate the calcium source associated with labetalol-induced endothelial calcium increases and labetalol-induced nitric oxide-mediated endothelial calcium regulation.

This study has some limitations. First, blood pressure is mainly affected by peripheral vascular resistance, which is determined by small resistance arterioles [26]. However, in this study, we used the aorta as a conduit vessel. Second, we used rat aortas for isometric tension measurements, whereas HUVECs were used to detect eNOS phosphorylation by western blotting. Third, this was an in vitro study; an in vivo study may be more appropriate for examining the effect of lipid emulsions on refractory cardiovascular depression caused by labetalol toxicity. Despite these limitations, the results suggest that the potency of labetalol in reducing blood pressure may be attenuated in patients with a compromised endothelium, such as those with atherosclerosis. As lipid emulsions (20% Intralipid), which were composed of 53% linoleic acid, 24% oleic acid, 11% palmitic acid, 8% alpha-linolenic acid, 4% stearic acid, and 0.1% arachidonic acid, reportedly alleviate refractory cardiovascular depression caused by a toxic dose of labetalol, they may be helpful in alleviating refractory cardiovascular depression caused by labetalol toxicity (toxic plasma concentration of labetalol: approximately 3.04 × 10^−6^ M) through inhibition of the labetalol-induced nitric oxide-mediated vasodilation [6,27,28]. However, further analyses, including case reports on the effect of lipid emulsions on refractory cardiovascular depression induced by toxic doses of labetalol, are needed. As 1% plasma triglyceride produces positive inotropic and scavenging effects on lipid-soluble drugs, a lipid emulsion dosing regimen to maintain 1% plasma triglyceride is suggested to alleviate the intractable cardiovascular depression caused by toxic doses of non-local anesthetic drugs [29,30]. Thus, the current study examined the effect of a 1% lipid emulsion on labetalol-induced vasodilation.

## 4. Materials and Methods

The experimental protocol (GNU-220420-R0040) was approved by the Institutional Animal Care and Use Committee of Gyeongsang National University, Republic of Korea. This study was conducted in accordance with the Animal Care and Use Guidelines of the Gyeongsang National University.

### 4.1. Preparation of Isolated Rat Aorta and Measurement of Isometric Tension

The thoracic aortas of the rats were prepared for tension assessment following a previously described method [31]. Male Sprague–Dawley rats weighing 240–280 g were anesthetized using 100% CO_2_ delivered through a small opening in the rat cage. Following extraction of the descending thoracic aorta, meticulous dissection under a microscope removed the surrounding adipose and connective tissues. Subsequently, the thoracic aorta was sectioned into small 2.5 mm segments. These segments were then affixed to a Grass isometric transducer (FT-03, Grass Instrument Co., Quincy, MA, USA) and submerged in a 10 mL organ bath containing Krebs solution maintained at 37 °C. The Krebs solution comprised sodium chloride (118 mM), sodium bicarbonate (25 mM), potassium chloride (4.7 mM), glucose (11 mM), calcium chloride (2.4 mM), monopotassium phosphate (1.2 mM), and magnesium sulfate (1.2 mM). To reach equilibrium, a baseline resting tension of 24.5 mN was maintained for 90 min, as per the method outlined by Klöss et al. [32]. Fresh Krebs solution was added every 30 min. Endothelium removal was performed to obtain the endothelium-denuded rat aorta, involving the insertion of two 25-gauge needles into the aortic lumen and manipulation of the aorta in a rolling motion around these needles. Endothelium removal from the isolated rat aortas was verified through the following method: Once a stable and sustained contraction was induced by 10^−8^ M phenylephrine, 10^−5^ M acetylcholine was introduced into the organ bath containing the endothelium-denuded isolated rat aorta, which exhibited the phenylephrine-induced contraction. If the relaxation induced by 10^−5^ M acetylcholine from the phenylephrine-induced contraction was < 15%, it was deemed indicative of the endothelium-denuded aorta. Endothelial integrity of the endothelium-intact aorta was validated as follows: Upon inducing a persistent and sustained contraction using phenylephrine (10^−7^ M), acetylcholine (10^−5^ M) was introduced into the organ bath containing the endothelium-intact aorta. If the relaxation induced by acetylcholine (10^−5^ M) from the phenylephrine-induced contraction exceeded 80%, the aorta was classified as the endothelium-intact aorta. Subsequently, both endothelium-intact and endothelium-denuded rat aortas, which displayed relaxation induced by acetylcholine, were repeatedly rinsed with fresh Krebs solution until they returned to their baseline resting tension.

### 4.2. Experimental Protocols

First, labetalol-induced vasodilation in endothelium-intact and endothelium-denuded rat aortas was examined. After phenylephrine (10^−6^ M) produced sustained and stable contractions in the endothelium-intact and endothelium-denuded rat aortas, labetalol (10^−7^ to 3 × 10^−4^ M) was cumulatively added to the organ bath to generate labetalol concentration–response curves in these aortas.

Second, the effects of various inhibitors on labetalol-induced vasodilation in the endothelium-intact aorta were examined to investigate the involvement of endothelial nitric oxide-mediated vasodilation and ATP-sensitive potassium channels in labetalol-induced vasodilation. After endothelium-intact aortas were pre-treated with the NOS inhibitor l-NAME (10^−4^ M), calmodulin-regulated enzyme inhibitor calmidazolium (3 × 10^−5^ M), non-specific GC inhibitor methylene blue (10^−6^ M), nitric oxide-sensitive GC inhibitor ODQ (10^−5^ M), and ATP-sensitive potassium channel inhibitor glibenclamide (5 × 10^−6^ M) for 20 min [33], phenylephrine (10^−6^ M) produced a sustained and stable contraction in endothelium-intact aortas in the presence or absence of various inhibitors. Then, labetalol (10^−7^ to 3 × 10^−4^ M) was cumulatively added to the organ bath to generate a labetalol concentration–response curve in endothelium-intact aortas in the presence or absence of various inhibitors.

Third, we examined the effects of various vasoconstrictors on labetalol-induced vasodilation in endothelium-denuded rat aortas. After various vasoconstrictors (10^−6^ M phenylephrine, 3 × 10^−6^ M 5-hydroxytryptamine, or 60 mM KCl) produced a stable and sustained contraction in the endothelium-denuded rat aortas, labetalol (10^−7^ to 3 × 10^−4^ M) was cumulatively added to the organ bath to generate a labetalol concentration–response curve.

Fourth, the effects of a lipid emulsion (Intralipid) on labetalol-induced vasodilation in endothelium-intact aortas with or without the NOS inhibitor l-NAME and endothelium-denuded aortas were examined to investigate the involvement of endothelial nitric oxide in the lipid emulsion-mediated change in labetalol-induced vasodilation. After pre-treating endothelium-intact or endothelium-denuded rat aortas with 1% lipid emulsions (Intralipid) for 20 min, phenylephrine (10^−6^ M) induced sustained and stable contractions in both types of aortas, regardless of the presence of the lipid emulsions. Then, labetalol (10^−7^ to 3 × 10^−4^ M) was cumulatively added to the organ bath to generate a labetalol concentration–response curve. Additionally, the effect of the NOS inhibitor l-NAME on lipid emulsion-mediated changes in labetalol-induced vasodilation in the endothelium-intact aorta was examined. Endothelium-intact aortas were pre-treated with l-NAME (10^−4^ M) alone for 35 min or with l-NAME (10^−4^ M) for 15 min, followed by lipid emulsions (1%) for 20 min. After phenylephrine (10^−6^ M) produced a stable and sustained contraction in l-NAME-pre-treated endothelium-intact aortas in the presence or absence of lipid emulsions (1%), labetalol (10^−7^ to 3 × 10^−4^ M) was cumulatively added to the organ bath to generate labetalol concentration–response curves.

### 4.3. cGMP Measurement

Isolated rat thoracic aortas were used to assess cGMP levels following the methodology outlined in a previous study [33]. cGMP levels were measured using the cGMP Complete Kit obtained from Abcam (Cambridge Science Park, Cambridge, UK). Endothelium-intact descending thoracic aortas were immersed in Krebs solution in a 10 mL organ bath for a duration of 60 min at 37 °C during drug treatment. The descending thoracic aorta containing the endothelium was treated with labetalol (3 × 10^−6^ M) for 10 min or lipid emulsions (1%) for 40 min. Additionally, endothelium-intact thoracic aortas were pre-exposed to l-NAME (10^−4^ M), ODQ (10^−5^ M) or lipid emulsions (1%) for 30 min before being treated with labetalol (3 × 10^−6^ M) for 10 min. Following treatment, the aortas were promptly frozen using liquid nitrogen and then homogenized in 0.1 M hydrochloride. The acidic supernatants were acetylated and cGMP levels were quantified using an enzyme-linked immunosorbent assay with the cGMP Complete kit. The concentration of cGMP obtained from each thoracic aorta was expressed as picomoles per milliliter (pmol/mL).

### 4.4. Cell Culture

HUVECs (C-0003-5C; American Type Culture Collection, Manassas, VA, USA) were cultured in an endothelial cell medium (ECM) obtained from ScienCell (Carlsbad, CA, USA). The ECM was supplemented with 15% fetal bovine serum (ScienCell), 100 units/mL penicillin, 1% endothelial cell growth supplement (ScienCell), and 100 μg/mL streptomycin (ScienCell), following the protocol described by Park et al. [33]. Cells at passages 3–5 were cultured in a serum-free ECM for 4 h before drug pre-treatment.

### 4.5. Western Blot

To investigate eNOS (at Ser1177 and Thr495) phosphorylation in HUVECs, western blotting was performed as previously described [33]. The cells underwent pre-treatment with either labetalol (3 × 10^−6^ M) alone for 5 min, lipid emulsions (1%) alone for 35 min, or lipid emulsions (1%) for 30 min followed by labetalol (3 × 10^−6^ M) for 5 min to assess eNOS phosphorylation at Ser1177 and Thr495. HUVECs were collected in radioimmunoprecipitation assay buffer (Cell Signaling Technology, Danvers, MA, USA), supplemented with phosphatase and protease inhibitor cocktails (Thermo Fisher Scientific, Rockfield, IL, USA). Following centrifugation at 20,000× *g* for 15 min at 4 °C, the protein content of the resulting supernatant was assessed using the bicinchoninic acid protein assay kit (Thermo Fisher Scientific). After boiling for 10 min, the samples containing 30 µg of protein were subjected to sodium dodecyl sulfate–polyacrylamide gel electrophoresis and subsequently transferred onto a polyvinylidene difluoride membrane (Millipore, Bedford, MA, USA). Then, the membranes were blocked using 5% skimmed milk in tris-buffered saline with 0.5% Tween-20 (TBST) for 60 min at 25 °C, followed by overnight incubation at 4 °C with primary antibodies (anti-phospho-eNOS at Ser1177 [diluted 1:1000], anti-phospho-eNOS at Thr495 [diluted 1:1000], anti-eNOS [diluted 1:1000], and anti-β actin [diluted 1:10,000]). Following the washes with TBST for 10 min each (repeated thrice), the membrane was treated with horseradish peroxidase-conjugated anti-rabbit or anti-mouse immunoglobulin G, diluted to 1:5000, for 60 min at 25 °C. Protein bands were visualized using the Westernbright^TM^ ECL western blotting detection kit (Advansta, Menlo Park, CA, USA). Protein band signals were imaged using a ChemiDoc^TM^ Touch Imaging System (Bio-Rad Laboratories, Inc., Hercules, CA, USA). Protein quantification was performed using ImageJ software (version 1.45s; National Institutes of Health, Bethesda, MD, USA).

### 4.6. Measurement of Intracellular Calcium Level

Intracellular calcium levels were assessed using a confocal laser microscope (IX70 Fluoview, Olympus, Tokyo, Japan) following a previously established protocol [33]. HUVECs were cultured in confocal cell culture dishes (SPL; Pocheon, Republic of Korea), treated with Fluo-4 AM (2.5 × 10^−6^ M, Invitrogen, Waltham, MA, USA) in Hank’s balanced salt solution medium for 30 min, and rinsed twice with phosphate-buffered saline solution. HUVECs were exposed to labetalol (3 × 10^−6^ M), l-NAME (10^−4^ M) and lipid emulsions (1%) individually. Additionally, the cells were treated with l-NAME (10^−4^ M) followed by labetalol (3 × 10^−6^ M) or lipid emulsions (1%) followed by labetalol (3 × 10^−6^ M). Subsequently, intracellular calcium levels were monitored at 2.5 s intervals using emission and excitation wavelengths of 520 and 485 nm, respectively. Fluo-4 AM-stained images were used to evaluate intracellular calcium levels, which were determined by dividing the fluorescence intensity (F) by the baseline fluorescence intensity (F_0_) measured prior to drug administration. The net alteration in calcium levels is represented by (F_max_ − F_0_)/F_0_, where F_max_ denotes the maximum calcium level determined from the fluorescence intensity following each treatment. Intracellular calcium levels were monitored for approximately 6 min.

### 4.7. Measurement of Labetalol Concentration in Distilled Water

Labetalol (10^−5^ M) was mixed with a lipid emulsion (Intralipid, 1%) in distilled water as per a previously described method [34]. The resulting emulsion was centrifuged at 18,600× *g* for 30 min to release labetalol. The concentration of labetalol in the aqueous phase was determined using ultra-performance liquid chromatography-quadrupole time-of-flight mass spectrometry (UPLC-Q-TOF MS; Waters, Milford, MA, USA). The aqueous layer was injected into an Acquity UPLC BEH C18 column (100 × 2.1 mm, 1.7 µm; Waters) equilibrated with a solvent mixture of water/acetonitrile (99:1) containing 0.1% formic acid (FA). Labetalol eluted by a linear gradient (1–100%) of acetonitrile containing 0.1% FA was analyzed by Q-TOF MS in positive electrospray ionization mode using the following optimal MS operating parameters: capillary voltage, 3 V; sampling cone voltage, 30 kV; source temperature, 400 °C; desolvation temperature, 100 °C; and desolvation flow rate, 800 L/h. The eluted labetalol was detected using the multiple reaction monitoring mode, with precursor and product ions identified by *m*/*z* 409.14 and 294.08, respectively. To ensure the accuracy and precision of all measurements, a lock spray procedure was performed using leucine–enkephalin ([M + H] = 556.2771) as the lock–mass reference. The MS data obtained were collected and analyzed using UIFI 1.8.2 software (Waters).

### 4.8. Materials

All chemicals used in this study were of the highest purity. Labetalol, l-NAME, calmidazolium, methylene blue, ODQ, 5-hydroxytryptamine, KCl, and phenylephrine were purchased from Sigma-Aldrich (St. Louis, MO, USA). Intralipid (20%) was obtained from Fresenius Kabi (Upsala, Sweden). Anti-phospho-eNOS (Ser1177 and Thr495) and anti-eNOS antibodies were purchased from Cell Signaling Technology and BD Biosciences (Franklin Lakes, NJ, USA), respectively. Calmidazolium and ODQ were solubilized in dimethyl sulfoxide (DMSO), with the final concentration of DMSO in the organ bath set at 0.1%. Unless otherwise specified, all remaining chemicals were dissolved in distilled water.

### 4.9. Statistical Analysis

The primary objective of this study was to examine the effects of endothelial denudation, various inhibitors, and lipid emulsions on labetalol-induced vasodilation. The effects of endothelial denudation, inhibitors, preconstrictors, and lipid emulsions on labetalol-induced vasodilation were analyzed using a linear mixed effects model (Stata version 14.2, Stata Corp LLC, Lakeway Drive, College Station, TX, USA) [35]. Log ED_50_ was calculated using non-linear regression by fitting the labetalol concentration–response curves to sigmoidal curves. This analysis was performed using Prism 5.0 (GraphPad Software Inc., San Diego, CA, USA). The effects of endothelial denudation and various inhibitors on the labetalol Log ED_50_ were analyzed using an unpaired Student’s *t*-test. Normality tests were performed using the Kolmogorov–Smirnov test. Areas under labetalol concentration–response curves were calculated using the trapezoid rule with Prism 5.0. Comparison of the difference in the area under the labetalol concentration–response curve in the presence or absence of 1% lipid emulsions between the endothelium-intact and endothelium-denuded aorta groups was performed using the unpaired Student’s *t*-test. Effect of the lipid emulsions on labetalol concentration were analyzed using an unpaired Student’s *t*-test. The effects of l-NAME, ODQ, and lipid emulsions on labetalol-induced cGMP formation or eNOS phosphorylation were analyzed using one-way analysis of variance followed by Bonferroni’s multiple comparison test. The effects of l-NAME and lipid emulsions on the labetalol-induced increase in intracellular endothelial calcium levels were analyzed using the Kruskal–Wallis test, followed by Dunn’s multiple comparison test. Values were considered significant at *p* < 0.05.

## 5. Conclusions

In conclusion, our findings suggest that lipid emulsions inhibit vasodilation induced by toxic doses of labetalol. This inhibition is mediated by the attenuation of labetalol-induced nitric oxide-dependent vasodilation and the absorption of labetalol.

## Figures and Tables

**Figure 1 ijms-25-07243-f001:**
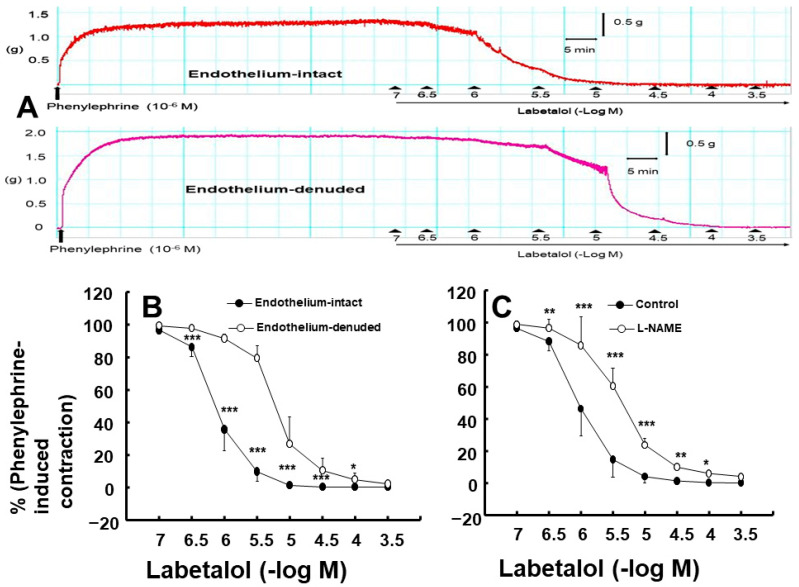
Labetalol-induced vasodilation in endothelium-intact and endothelium-denuded rat aortas. (**A**) Original tracing showing labetalol (10^−7^ to 3 × 10^−4^ M)-induced vasodilation in the endothelium-intact and endothelium-denuded rat aortas pre-contracted with phenylephrine (10^−6^ M). (**B**) Effect of endothelial denudation (N = 7) on labetalol-induced vasodilation in isolated rat aortas. (**C**) Effect of N^W^-nitro-l-arginine methyl ester (l-NAME, 10^−4^ M, N = 7) on labetalol-induced vasodilation in endothelium-intact rat aortas. Data were analyzed using a linear mixed effects model. Data are shown as means ± standard deviations and expressed as percentages of contraction induced by phenylephrine (10^−6^ M). N indicates the number of rats from which isolated aortas were obtained. * *p* < 0.05, ** *p* < 0.01, and *** *p* < 0.001 vs. endothelium-denuded or control.

**Figure 2 ijms-25-07243-f002:**
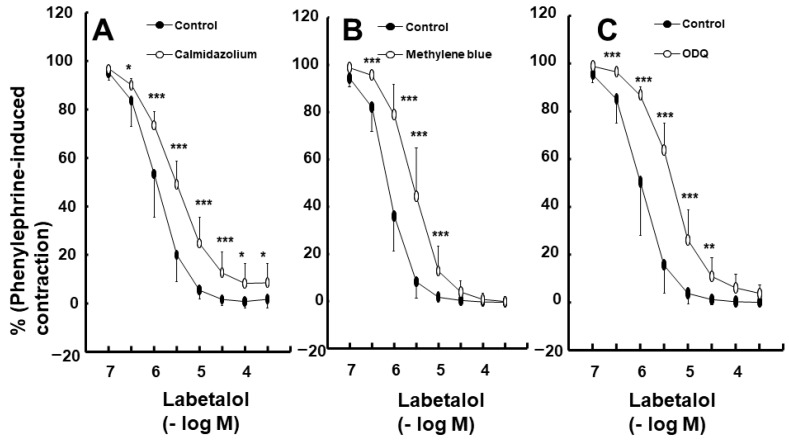
Effect of calmidazolium (3 × 10^−5^ M, N = 6) (**A**), methylene blue (10^−6^ M, N = 8) (**B**), and ODQ (10^−5^ M, N = 7) (**C**) on labetalol-induced vasodilation in endothelium-intact rat aortas. Data were analyzed using a linear mixed effects model. Data are shown as means ± standard deviations and expressed as percentages of contraction induced by phenylephrine (10^−6^ M). N indicates the number of rats from which isolated aortas were obtained. * *p* < 0.05, ** *p* < 0.01, and *** *p* < 0.001 vs. control.

**Figure 3 ijms-25-07243-f003:**
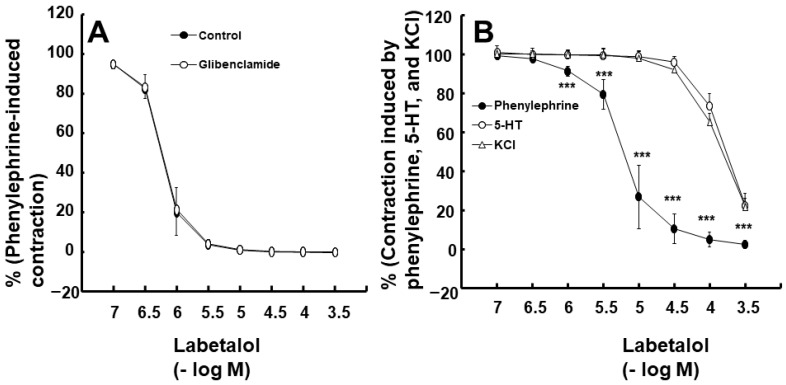
(**A**) Effect of glibenclamide (5 × 10^−6^ M) on labetalol-induced vasodilation in endothelium-intact rat aortas. Data were analyzed using a linear mixed effects model. Data (N = 5) are shown as means ± standard deviations and expressed as percentages of contraction induced by phenylephrine (10^−6^ M). N indicates the number of rats from which isolated aortas were obtained. (**B**) Effects of phenylephrine (10^−6^ M, N = 7), 60 mM KCl (N = 6), and 5-hydroxytryptamine (5-HT, 3 × 10^−6^ M, N = 6) on labetalol-induced vasodilation in endothelium-denuded aortas. Data were analyzed using a linear mixed effects model, presented as means ± standard deviations, and expressed as percentages of contraction induced by phenylephrine, KCl, and 5-HT. N indicates the number of rats from which isolated aortas were obtained. *** *p* < 0.001 vs. KCl or 5-HT.

**Figure 4 ijms-25-07243-f004:**
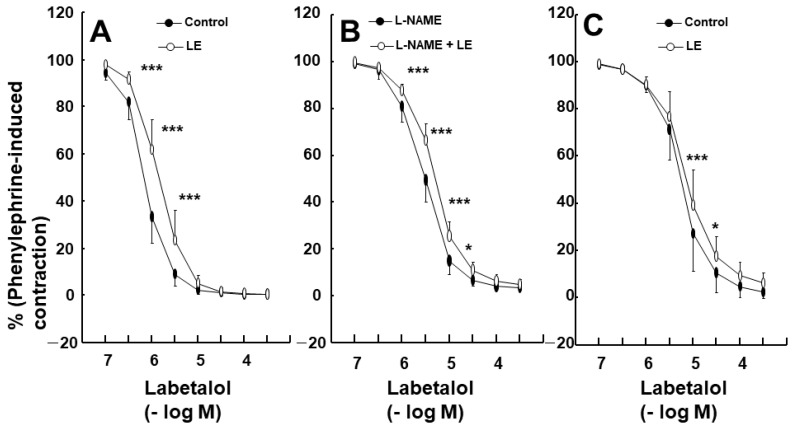
Effect of lipid emulsion (LEs, 1%) on labetalol-induced vasodilation in endothelium-intact rat aortas, without (N = 8) (**A**) or with (**B**) (N = 5) N^W^-nitro-l-arginine methyl ester (l-NAME, 10^−4^ M), and (**C**) in endothelium-denuded rat aortas (N = 7). Data were analyzed using a linear mixed effects model. Data are shown as means ± standard deviations and expressed as percentages of contraction induced by phenylephrine (10^−6^ M). N indicates the number of rats from which isolated aortas were obtained. * *p* < 0.05 and *** *p* < 0.001 vs. control or l-NAME alone.

**Figure 5 ijms-25-07243-f005:**
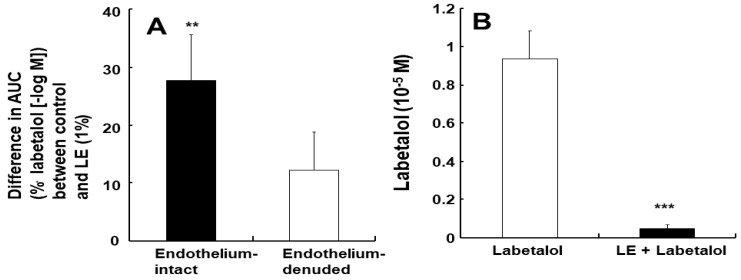
(**A**) Effect of endothelial denudation on the difference in area under curve (AUC) with labetalol-induced vasodilation between control and lipid emulsions (LEs, 1%) (endothelium-intact: N = 8; endothelium-denuded: N = 7) in isolated rat aortas. N indicates the number of rats from which isolated aortas were obtained. ** *p* < 0.01 vs. endothelium-denuded. (**B**) Effect of LEs (1%) on labetalol (10^−5^ M) concentration in distilled water. Labetalol (10^−5^ M) released from a 1% Intralipid emulsion in distilled water was analyzed using ultra-performance liquid chromatography-quadrupole time-of-flight mass spectrometry with positive electrospray ionization in multiple reaction monitoring mode. Data were analyzed using the unpaired Student’s *t*-test, and are presented as means ± standard deviations. The experiment was repeated eight times. *** *p* < 0.001 vs. labetalol alone.

**Figure 6 ijms-25-07243-f006:**
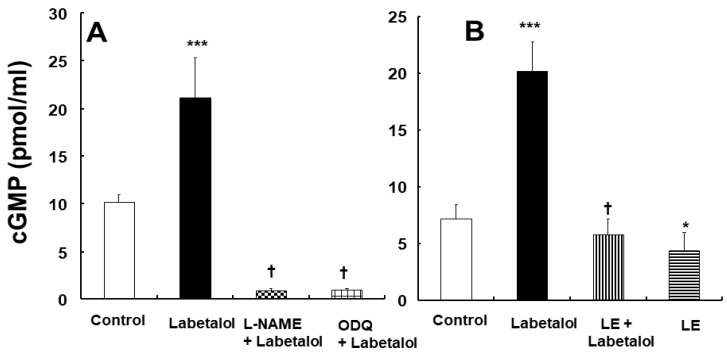
Effect of labetalol (3 × 10^−6^ M; (**A**): N = 5; (**B**): N = 6), N^W^-nitro-l-arginine methyl ester (l-NAME, 10^−4^ M, N = 5, (**A**)), 1*H*-[1,2,4]oxadiazolo[4,3-a]quinoxalin-1-one (ODQ, 10^−5^ M, N = 5, (**A**)), and lipid emulsions (LEs, 1%, N = 6, (**B**)), alone or combined, on the cyclic guanosine monophosphate (cGMP) formation in isolated endothelium-intact rat aorta. N indicates the number of rats. Data were analyzed using one-way analysis of variance, followed by Bonferroni’s multiple comparison tests. Data are shown as means ± standard deviations. * *p* < 0.05 and *** *p* < 0.001 vs. control. † *p* < 0.001 vs. labetalol alone.

**Figure 7 ijms-25-07243-f007:**
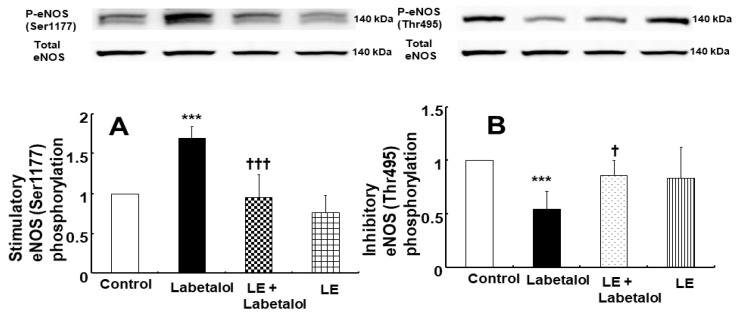
Effect of labetalol (3 × 10^−6^ M) and lipid emulsions (LEs, 1%), alone or combined, on endothelial nitric oxide synthase (eNOS, Ser1177 (**A**) and Thr495 (**B**)) phosphorylation in human umbilical vein endothelial cells. Data were analyzed using one-way analysis of variance, followed by Bonferroni’s multiple comparison tests. Data (N = 4) are shown as means ± standard deviations. N indicates the number of independent experiments. *** *p* < 0.001 vs. control. † *p* < 0.05 and ††† < 0.001 vs. labetalol alone. P-eNOS, phosphorylated eNOS.

**Figure 8 ijms-25-07243-f008:**
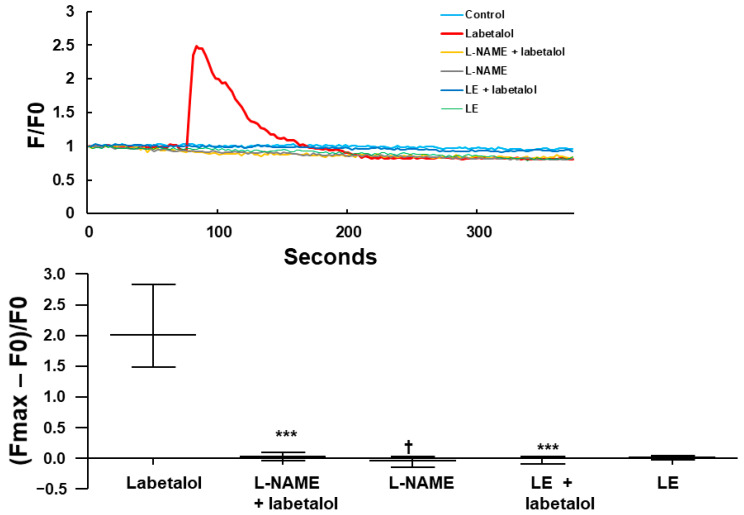
Effect of labetalol (3 × 10^−6^ M), N^W^-nitro-l-arginine methyl ester (l-NAME, 10^−4^ M), and lipid emulsions (LEs, 1%), alone or combined, on intracellular calcium level in human umbilical vein endothelial cells. Data were analyzed using the Kruskal–Wallis test, followed by Dunn’s multiple comparison test. Data (N = 6) are shown as medians ± interquartile ranges (25% to 75%). N indicates the number of independent experiments. *** *p* < 0.001 vs. labetalol. † *p* < 0.05 vs. l-NAME + labetalol.

## Data Availability

The data presented in this study are available on reasonable request from the corresponding author.

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
