# Peer review of "Lipid Emulsions Inhibit Labetalol-Induced Vasodilation in the Isolated Rat Aorta"

_ijms, 2024, doi:10.3390/ijms25137243_

Round 1

Reviewer 1 Report

Comments and Suggestions for Authors

Present study in animal experimental model of SD rats and HUVEC cell line examined the effect and the mechanisms of labetolol induced vasodilation  (in relation to NO and NOS) and the effect of lipid emulsion in labetolol toxicity in vitro. Results suggest (contrary to one study performed in human samples of radial arteries) that labetolol exerts its effect via NOS pathway, Additionally, LE is protective against labetolol toxicity.

Comments and suggestions:

Materials and methods: How was the toxic dose for cell cultures determined and validated? I found the toxic dose for labetolol inducing vasodilation in the manuscript. Was viability of the cells evaluated? How is toxicity of labetolol in the HUVEC assessed?

Results: Please add which statistical test was used for comparisons in the figure legends

Conclusion: please remove the second sentence, since experimental design (in vitro, without measurements of arterial blood pressure) does not allow to conclude that. Results are suggestive and one may speculate, but to conclude about pressure, one must to perform experiments in animal models or humans.

Author Response

Responses to Reviewer #1’s comments

Thank you very much for thoughtful comments. 

  • Materials and methods: How was the toxic dose for cell cultures determined and validated? I found the toxic dose for labetolol inducing vasodilation in the manuscript. Was viability of the cells evaluated? How is toxicity of labetolol in the HUVEC assessed?

Response:

We would like to thank the reviewer for evaluating our manuscript and providing insightful comments. To detect endothelial nitric oxide synthase (eNOS) phosphorylation in HUVECs, we used a labetalol concentration (labetalol: 3 × 10-6 M) that was determined to induce the maximum difference in labetalol-induced vasodilation between the endothelium-intact and the endothelium-denuded aorta. In addition, we considered toxic plasma concentration of labetalol (approximately 3.04 × 10-6 M) in the experiment regarding the labetalol-induced eNOS phosphorylation. Thus, we used 3 × 10-6 M labetalol to detect eNOS phosphorylation in western blot. In the experiment assessing eNOS phosphorylation, HUVECs were pre-treated as follows: labetalol (3 × 10-6 M) alone for 5 min, lipid emulsion (1%) for 30 min followed by labetalol (3 × 10-6 M) for 5 min, or lipid emulsion (1%) alone for 35 min. This incubation period of less than 1 h and the relatively low concentration were chosen to minimize any potential impact on HUVEC viability.

  • Results: Please add which statistical test was used for comparisons in the figure legends

Response:

We have provided this information into each figure legend of the revised manuscript, following this reviewer suggestion.

  • Conclusion: please remove the second sentence, since experimental design (in vitro, without measurements of arterial blood pressure) does not allow to conclude that. Results are suggestive and one may speculate, but to conclude about pressure, one must to perform experiments in animal models or humans.

Response:

We have removed the following sentence from the Conclusion section of revised manuscript. “Thus, lipid emulsions may help to alleviate refractory hypotension due to severe vasodilation caused by toxic dose of labetalol administration.”

Reviewer 2 Report

Comments and Suggestions for Authors

The manuscript “ Lipid Emulsions Inhibit Labetalol-Induced Vasodilation in Isolated Rat Aorta” by Hee Lee et al is a very interesting report on Labetalol-Induced endothelium-dependent vasodilation. However, the experimental setup is inadequate even when the results are good.

Several relevant references were not included, as an example: Mostaghim et al (J Pharmacol & Exp Therapeutics 239 (3): 1986) showed that labetalol relaxed endothelium intact and denuded vessels and only when phenylephrine was used as pre-constrictor, suggesting a displacement from alpha-adrenergic receptors

An IC50 comparison among different treatments will be helpful including confidence ranges and statistical significance

Are control curves similar in endothelium intact vessels (Fig 1B and Fig 4)? And endothelium-denuded (Figure 1B and Figure 4C)?

Are the IC50 different between control and LE treatments (Figure 4 C)? .Do the authors

How was the AUC calculated from 0 or 100 (Figure 5 A)? Please explain

Explain how (Figure 5B) a 9 times lower concentration of labetalol induced similar dilation levels?

Line 182-185 .

“Labetalol (3 × 10-6 M) increased the intracellular calcium levels in HUVECs (Fig. 8).  However, pre-treatment with the L-NAME (10-4 M) or lipid emulsion (1%) attenuated the  increase in calcium levels induced by 3 × 10-6 M labetalol in HUVECs (Fig. 8; P < 0.001  versus labetalol alone).”

I do not understand why L-NAME, an inhibitor of NOS, can block the intracellular calcium kinetics induced by an adrenergic receptor agonist. Are the authors proposing an interaction with the intracellular pathways stimulated by Labetalol's interaction with the adrenergic receptors?

This fact needs a very deep discussion.

The discussion is out of focus

Line 192

“This study suggests that lipid emulsion reverses vasodilation induced by a toxic dose of labetalol in isolated rat aortas”. What do the authors mean by reverses? Block?, change direction? If a  3.04 X 10-6M concentration is considered a toxic concentration, explain/discuss why (Figure 4C) there are no differences from 10-7-10-5.5 M. Why is the blockage not happening at a lower concentration?

What is the composition of lipid emulsion) (intralipid)?

Comments on the Quality of English Language

none

Author Response

Responses to Reviewer #2’s comments

Thank you very much for thoughtful comments. 

  • Several relevant references were not included, as an example: Mostaghim et al (J Pharmacol & Exp Therapeutics 239 (3): 1986) showed that labetalol relaxed endothelium intact and denuded vessels and only when phenylephrine was used as pre-constrictor, suggesting a displacement from alpha-adrenergic receptors

Response:

We would like to thank the reviewer for evaluating our manuscript and providing insightful comments. Please note that we have cited this article in the Introduction section of the revised manuscript. The added sentence is as follows: “Additionally, labetalol elicits greater relaxation in endothelium-intact aorta compared to in endothelium-denuded aorta, suggesting that labetalol-induced vasodilation is endothelium-dependent [4].”

  • An IC50 comparison among different treatments will be helpful including confidence ranges and statistical significance

Response:

As the reviewer recommended, log ED50 was calculated using non-linear regression by fitting labetalol concentration-response curves to a sigmoidal curves using Prism 5.0 (GraphPad Software Inc., San Diego, CA, USA). In addition, the effect of various inhibitors and endothelial denudation on the labetalol log ED50 was analyzed using unpaired Student’s t-test. These ED50 data were added into the Results section of the revised manuscript. However, statistical analysis adopting linear mixed effect model can be used to analyze the effect of endothelial denudation and various inhibitors on the labetalol-induced vasodilation at each labetalol concentration (Comput Stat. 2008, 23, 99–109). Thus, in the revised manuscript, we used two statistical methods to analyze the effects of endothelial denudation and various inhibitors on labetalol-induced vasodilation.

  • Are control curves similar in endothelium intact vessels (Fig 1B and Fig 4)? And endothelium-denuded (Figure 1B and Figure 4C)?

Response:

Labetalol log ED50 values in the endothelium-intact aorta were -6.11 ± 0.11 (Figure 1b) and -6.14 ± 0.11 (Figure 4a), indicating a similar labetalol log ED50 in the endothelium-intact control aorta across both figures. Similarly, in the endothelium-denuded aorta, labetalol log ED50 was -5.20 ± 0.15 (Figure 1b) and -5.25 ± 0.19 (Figure 4c), also showing comparable labetalol log ED50 in the endothelium-denuded control aorta across these figures. Therefore, the control curves of both endothelium-intact (Figures 1b and 4a) and endothelium-denuded (Figures 1b and 4c) aortas appear consistent.

  • Are the IC50 different between control and LE treatments (Figure 4 C)? .Do the authors

Response:

As suggested by the reviewer, we compared the labetalol log ED50 values for labetalol-induced vasodilation between the control and lipid emulsion groups in the endothelium-denuded aorta (Fig. 4C). The labetalol log ED50 of control and LE was -5.25 ± 0.19 and -5.15 ± 0.18, respectively, which is not statistically significant (P = 0.17). However, the effects of lipid emulsion on labetalol-induced vasodilation at each concentration of labetalol in the endothelium-intact and -denuded aortas were analyzed using only linear mixed effects model in the current study because linear mixed effects model produced significant difference of labetalol-induced vasodilation between control and lipid emulsion in endothelium-denuded aorta . Thus, we used results (Figure 4a, b and c) obtained from statistical analysis using only the linear mixed effects model in the revised manuscript.

  • How was the AUC calculated from 0 or 100 (Figure 5 A)? Please explain

Response:

AUC is calculated using trapezoid rules with Prism 5.0. The following sentence was added into the “Statistical analysis” subsection of the revised manuscript to indicate the same: “Area under labetalol concentration-response curves was calculated using trapezoid rule with Prism 5.0.”

  • Explain how (Figure 5B) a 9 times lower concentration of labetalol induced similar dilation levels?

Response:

High-performance liquid chromatography was performed to examine the effect of lipid emulsion on the labetalol concentration in distilled water. However, isometric tension measurement was performed in Krebs solution, which contains sodium chloride (118 mM), sodium bicarbonate (25 mM), potassium chloride (4.7 mM), glucose (11 mM), calcium chloride (2.4 mM), monopotassium phosphate (1.2 mM), and magnesium sulfate (1.2 mM). This discrepancy between the results of HPLC and isometric tension measurement could be attributed to the different ionic strength (calculated ionic strength of Krebs solution: 158 mM) (J Chromatogr A. 2004 Apr 9;1033(1):57-69). This discrepancy is difficult to explain. We hypothesize that lipid emulsion-mediated inhibition of nitric oxide release and lipid emulsion-mediated sequestration of labetalol inhibited the labetalol-induced vasodilation observed at lower concentrations of labetalol in the endothelium-intact aorta (Figure 4a). Moreover, lipid emulsion-mediated sequestration of labetalol (10-5 M) attenuated labetalol-induced vasodilation at relatively high concentrations of labetalol in the endothelium-denuded aorta (Figure 4c).

  • Line 182-185 .

“Labetalol (3 × 10-6 M) increased the intracellular calcium levels in HUVECs (Fig. 8).  However, pre-treatment with  the L-NAME (10-4 M) or lipid emulsion (1%) attenuated the  increase in calcium levels induced by 3 × 10-6 M labetalol in HUVECs (Fig. 8; P < 0.001  versus labetalol alone).”

I do not understand why L-NAME, an inhibitor of NOS, can block the intracellular calcium kinetics induced by an adrenergic receptor agonist. Are the authors proposing an interaction with the intracellular pathways stimulated by Labetalol's interaction with the adrenergic receptors?

This fact needs a very deep discussion.

Response:

Endothelial calcium increase is essential to activate eNOS. Pre-treatment with L-NAME inhibited endothelial calcium increase induced by chloroquine in HUVECs, leading to a decrease in nitric oxide production. Nitric oxide decreases endothelial calcium level via a cGMP-dependent mechanism, whereas it increases endothelial calcium level via a cGMP-independent mechanism. The following part was added into the Discussion section of the revised manuscript: “Acetylcholine produces intracellular calcium oscillations associated with nitric oxide release in endothelial cells, which is induced by endogenous calcium release from the endoplasmic reticulum and calcium entry via store-operated calcium channels [23]. Nitric inhibits endothelial calcium level via a cGMP-dependent mechanism, but also increases endothelial calcium through a cGMP-independent mechanism [24]. Moreover, nitric oxide decreases endothelial calcium level by attenuating capacitative calcium entry and enhancing calcium uptake into the endoplasmic reticulum via a cGMP-dependent mechanism [25]. This contributes to the autoregulation of nitric oxide production [25]. Further studies are needed to investigate the calcium source associated with labetalol-induced endothelial calcium increase and labetalol-induced nitric oxide-mediated endothelial calcium regulation.”

  • The discussion is out of focus

Line 192

“This study suggests that lipid emulsion reverses vasodilation induced by a toxic dose of labetalol in isolated rat aortas”. What do the authors mean by reverses? Block?, change direction? If a  3.04 X 10-6M concentration is considered a toxic concentration, explain/discuss why (Figure 4C) there are no differences from 10-7-10-5.5 M. Why is the blockage not happening at a lower concentration?

Response:

This sentence was slightly modified using the word “attenuates.” The modified sentence is as follows: “This study suggests that lipid emulsion attenuates vasodilation induced by a toxic dose of labetalol in isolated rat aortas, mediated through inhibition of nitric oxide release and reduction of labetalol concentration.”

It is very difficult to explain the lack of significant difference of labetalol (10-7 to 3 × 10-6 M)-induced vasodilation between control and lipid emulsion in the endothelium-denuded aorta (Figure 4c). However, lipid emulsion inhibited labetalol (3 × 10-7 to 3 × 10-6 M)-induced vasodilation at lower concentrations of labetalol in endothelium-intact aorta. This finding could be attributed to both lipid emulsion-induced inhibition of labetalol-induced nitric oxide release and lipid emulsion-mediated sequestration of labetalol, which appears to be effective at low concentrations of labetalol (Figure 4a). Moreover, lipid emulsion inhibited labetalol-induced vasodilation at relatively high concentrations of labetalol (10-5 and 3 × 10-5 M) in the endothelium-denuded aorta (Figure 4c), which can be attributed to lipid emulsion-mediated sequestration of labetalol. We have added this information to the Discussion section of the revised manuscript.

  • What is the composition of lipid emulsion) (intralipid)?

Response:

We have added the composition of lipid emulsion into the Discussion section of the revised manuscript. The added part is as follows:

“As lipid emulsion (20% Intralipid), which is composed of 53% linoleic acid, 24% oleic acid, 11% palmitic acid, 8% alpha-linolenic acid, 4% stearic acid, and 0.1% arachidonic acid, reportedly alleviates refractory cardiovascular depression caused by a toxic dose of labetalol, it may be helpful in alleviating refractory cardiovascular depression caused by labetalol toxicity (toxic plasma concentration of labetalol: approximately 3.04 × 10-6 M) through inhibition of the labetalol-induced nitric oxide-mediated vasodilation [6,27,28].”

Reviewer 3 Report

Comments and Suggestions for Authors

The study titled "Lipid Emulsions Inhibit Labetalol-Induced Vasodilation in Isolated Rat Aorta" by Soo Hee Lee et al., is interesting and novel. The use of lipid emulsion to alleviate labetalol-induced toxicity in both endothelium-intact and denuded rat aorta. The study design is appropriate, and the results support the conclusion. I have only few questions and suggestions to better understand the effects of Labetalol on cells and how lipid emulsions alleviate this.

1) In Figure 8, the calcium spikes seen are intracellular calcium or extracellular calcium uptake in response to labetalol? It would be interesting to understand if the HUVEC cells induced by labetalol undergo apoptosis following the calcium overload and whether the lipid emulsions prevent this. 

2) Does pre-treatment of the HUVEC cells or aorta with these chemicals in any way affect the viability of the cells? Have you tested the doses of the chemicals for dosage and viability response?

Author Response

Responses to Reviewer #3’s comments

Thank you very for thoughtful comments. 

  • In Figure 8, the calcium spikes seen are intracellular calcium or extracellular calcium uptake in response to labetalol? It would be interesting to understand if the HUVEC cells induced by labetalol undergo apoptosis following the calcium overload and whether the lipid emulsions prevent this.

Response:

We would like to thank the reviewer for evaluating our manuscript and providing insightful comments. Since we did not examine the source of calcium, the current experiment does not clarify whether the labetalol-induced increase in calcium is caused by the release from the endoplasmic reticulum or store-operated calcium entry. We have added the following part to the Discussion section of the revised manuscript.

“Acetylcholine produces intracellular calcium oscillations associated with nitric oxide release in endothelial cells, which is induced by endogenous calcium release from the endoplasmic reticulum and calcium entry via store-operated calcium channels [23]. Nitric oxide inhibits endothelial calcium level via a cGMP-dependent mechanism, but also increases endothelial calcium through a cGMP-independent mechanism [24]. Moreover, nitric oxide decreases endothelial calcium level by attenuating capacitative calcium entry and enhancing calcium uptake into the endoplasmic reticulum via a cGMP-dependent mechanism [25]. This contributes to the autoregulation of nitric oxide production [25]. Further studies are needed to investigate the calcium source associated with labetalol-induced endothelial calcium increase and labetalol-induced nitric oxide-mediated endothelial calcium regulation.”

  • Does pre-treatment of the HUVEC cells or aorta with these chemicals in any way affect the viability of the cells? Have you tested the doses of the chemicals for dosage and viability response?

Response:

The HUVECs used in the western blotting experiment to detect eNOS phosphorylation were pre-treated with labetalol (3 × 10-6 M: approximately equivalent to minimal toxic plasma concentration of labetalol) alone for 5 min, lipid emulsion (1%) for 30 min followed by labetalol (3 × 10-6 M) for 5 min, or lipid emulsion (1%) alone for 35 min, which is less than 1 h. Considering this short incubation time and concentration, there appears to be no adverse effect on the viability of HUVECs. Moreover, the concentration of labetalol used for labetalol concentration-response curves and concentration of chemical (inhibitors) to be used for pre-treatment in the rat aorta were selected based on previous relevant articles (J Thorac Cardiovasc Surg. 2015, 149, 1036-40; Gen Physiol Biophys 2023, 42: 469-478).

Round 2

Reviewer 3 Report

Comments and Suggestions for Authors

The responses to the comments are satisfactory and the addition of the discussion section on calcium in the revised manuscript is appropriate.